# Evaluation of Selective Culling as a Containment Strategy for African Swine Fever at a Vietnamese Sow Farm

**DOI:** 10.3390/pathogens13070567

**Published:** 2024-07-06

**Authors:** Bui Thi To Nga, Agathe Auer, Pawin Padungtod, Klaas Dietze, Anja Globig, Andriy Rozstalnyy, Tran Minh Hai, Klaus Depner

**Affiliations:** 1Faculty of Veterinary Medicine, Vietnam National University of Agriculture, Hanoi 100000, Vietnam; 2Food and Agriculture Organization of the United Nations (FAO), 00153 Rome, Italy; 3Food and Agriculture Organization of the United Nations (FAO), Representation in Vietnam, Hanoi 100000, Vietnam; 4Friedrich-Loeffler-Institut, 17493 Greifswald-Insel Riems, Germany

**Keywords:** disease control, culling strategy, early detection, slow contagion

## Abstract

Selective culling, also known as the “tooth extraction approach”, is a strategy for controlling African swine fever (ASF) by removing only sick and suspect animals instead of the entire herd in Vietnam. This method prioritizes preserving healthy animals, particularly valuable breeding pigs. Despite its implementation in various forms, no standardized protocol based on scientific principles has been established. Farms typically adapt this strategy based on their understanding, which can vary significantly. In implementing of selective culling that is not based on scientific principles, there is a significant risk of spreading the disease. The aim of this study is to evaluate the consequences of selective culling as currently implemented in Vietnam. Our analysis on a large sow farm revealed that current practices rely heavily on clinical observations without laboratory confirmations. This approach allows ASF-infected animals to remain on the farm longer, potentially exacerbating the spread of the virus. Thus, selective culling poses a substantial risk by potentially exacerbating the spread of disease. Our findings emphasize that early diagnosis of ASF and systematic removal of infected pigs are critical components for the effective implementation of selective culling strategies and that a high level of fragmentation to minimize contact between animals plays a key role. The optimal approach is to test conspicuous animals and separate them. Under no circumstances should suspect animals be left in the herd for several days before they become severely ill and succumb to the disease.

## 1. Introduction

African swine fever (ASF) is a fatal hemorrhagic disease that affects susceptible species within the *Suidae* family, including domestic and wild pigs. This disease causes severe economic losses and substantially impacts the livelihoods of pig producers across various production systems [1,2,3,4]. The incursion of ASF into the Asia–Pacific region has been a significant concern, particularly as the region hosts more than half of the global pig population [2]. In Asia and the Pacific, a total of 19 countries have reported ASF outbreaks as of May 2024 [5]. The current focus of regional disease managers is to shift from addressing ASF as an emerging disease with the aim of elimination to managing ASF as an endemic disease. This involves a paradigm shift towards minimizing the ongoing impacts on food security and endemic stability [6,7,8,9,10,11,12].

In Vietnam, pig production accounts for approximately 60% of the total livestock-related output and provides livelihoods for nearly three million households [13,14]. The majority of farms are backyard farms, with an average herd size ranging from 4 to 10 pigs per household, collectively contributing to around 80% of the country’s pig production [15,16]. The sector significantly supports both domestic and export markets [17,18]. However, despite its importance, the sector faces various challenges, including the persistent threat of disease outbreaks and price volatility, particularly impacting smallholder producers [12,19].

Since 2019, Vietnam’s pig industry has faced a significant challenge due to ASF outbreaks across the country. From February 2019 to February 2020, ASF outbreaks occurred in 8548 communes in 667 districts of 63 provinces and cities, requiring the culling of approximately six million pigs [20,21]. This severe loss of animals represented a 25% decrease in the total pig population by December 2019 compared to 2018 [5]. In July 2019, the Ministry of Agriculture and Rural Development amended the ASF control policy to include the possibility of partial and selective culling on infected farms (official letter 5169/BNN-TY). This policy allows the disposal of only dead, sick, and ASF-positive pigs in outbreak farms [22]. Healthy pigs testing negative for ASF may be slaughtered within the outbreak area. This form of selective culling, in which only the clinically sick and ASF-positive pigs are removed, is also casually addressed as the “tooth extraction” strategy. A previous study in Vietnam based on field data showed that the selective culling approach resulted in an average survival rate of about 54% of the total herd, with no significant effect on the overall duration of the control intervention [9]. Evidence was provided to underline the positive effects of the resource-saving partial culling approach without compromising the time effectiveness of control interventions.

However, a standardized protocol for selective culling based on scientific considerations does not exist. Affected farms usually implement the “tooth extraction” strategy according to their own understanding.

In this short communication, we present data on the selective culling strategy implemented on a large, ASF-affected sow farm in Vietnam under local circumstances. We aimed to identify weaknesses, evaluate consequences, and suggest how the selective culling measures could be optimized or improved under sub-optimal field conditions. Finally, our recommendations could facilitate drafting a standardized protocol for selective culling to be used by farmers throughout the country.

## 2. Materials and Methods

### 2.1. Farm Description

The farm in question is a sow breeding farm with a capacity of 1200 sows. It is divided into two sections, A and B, which are approximately 50 m apart and located within a fenced area. There is no animal movement between sections, except for the initial quarantine period. Both sections manage all production phases independently in a multi-site production system.

To provide a clearer understanding of the farm’s layout and structure, Figure 1 presents a schematic view of the farm, detailing the placement of each section and stable. 

Section A consists of five stables, while section B consists of four stables. Each section of the farm (A and B) operates as an independent epidemiological unit. Animals in different units do not mix, except during the initial quarantine period, to minimize the risk of disease spread. In addition to the sows, 18 boars were kept for semen collection, with 10 located in section A and 8 in section B. There are two stables designated for pregnant sows and six for farrowing. Pregnant sows and those to be inseminated are kept individually in cages, while after farrowing, they are housed in group stables with their piglets. The piglets remain with the sow for 21 days before being weaned and moved to another facility for fattening.

One of the stables in section A is used exclusively for quarantine. The farm regularly buys gilts to restock. Newly purchased animals are housed here for about three weeks before being moved to other stables. Sows are moved between stables within one section according to their production status and, under certain circumstances, between the two sections. This was the case during the first wave of infection in 2022 when section A was completely emptied. Biosecurity measures were implemented, including a vehicle tire bath prior to entry, mandatory showers for personnel and visitors, a three-day quarantine for visitors, appropriate personal protective equipment, and the use of decontamination chemicals.

### 2.2. ASF History

Information on ASF on the farm was provided by the farm veterinarian and employees. The ASF events occurred in two waves, both confirmed by real-time PCR (qPCR). The timeline of the infection is shown in Figure 2. The first wave of infection lasted from October 2022 to February 2023 and mainly affected section A, where a total of 730 out of 800 animals had to be removed, either because they died, became sick, or were in direct contact with infected animals. During the first week of infection, approximately 20 sows per day had to be removed and culled, followed by approximately 10 sows per day for the next 2 weeks, and then approximately 5 sows per day. After ASF was initially confirmed, the removal of animals was based on clinical signs. No selective culling was performed. In section B, only two animals fell ill and had to be culled. Finally, 70 remaining clinically healthy pigs from section A were moved to section B in February 2023. The stables in section A were then kept empty until July 2023, when 600 new sows were introduced. 

In February 2024, the second ASF wave started, affecting both sections A and B. The infection is currently ongoing (as of May 2024). To control the disease, selective culling was implemented as follows: Clinically ill sows (F0) were first isolated and treated with antibiotics. The authors emphasize that antibiotics are not intended to prevent the spread of viral diseases, and such practices should not be implemented on farms infected with African Swine Fever (ASF). However, this was observed to be the practice on the farm visited. If there was no recovery within 3 to 5 days and the clinical signs worsened, the animals were culled. Animals in direct contact (F1) with F0 were also removed and isolated in the same barn, while animals with secondary contact (F2) remained in their original locations (Figure 3). Sick animals were driven to a collection point via a lane covered with large canvases dusted with lime powder. After use, these canvases were burned, and the lane was cleaned and disinfected. Deceased pigs were transported by cart to an incineration place on the farm. Additionally, to reduce the spread of the virus among the animals, the housing of the sows was altered so that three stalls were occupied by sows, followed by an empty stall (Figure 3). This change led to a reduction in the total number of sows to 1000.

Up to 5 animals per day have died or been culled, totaling 100 from February to April 2024 in both sections A and B, which represents a loss rate of 10% of the herd over three months. A comparison of the survival rate dynamics during the two detected ASF waves on the farm is depicted in Figure 4. Figure 4 was created using GraphPad Prism version 10.2.1 for Windows, GraphPad Software, Boston, MA, USA, www.graphpad.com. So far, about 50% of the F1 animals have become infected and either died or been culled, while none of the F2 sows have become ill. Sick sows were defined as having ASF if they had not eaten for several days and were lying down, unable to get up or had a fever. In cases of milder signs like loss of appetite, animals were first treated with antibiotics to prevent secondary infections, although it is known that this does not combat ASF itself. Culling was based solely on clinical signs since ASF-specific laboratory tests were not conducted.

### 2.3. Pen-Side Tests

To assess the effectiveness of selective culling practices implemented on the farm, a detailed evaluation was conducted during a scheduled visit. On the day of the visit, the farm did not report any severely ill animals. However, observations were made of three pregnant sows, which, despite having not consumed food for several days, remained mobile but lethargic, standing up only when forced. Notably, these sows were not housed next to each other and showed no signs of hemorrhagic skin lesions, which are commonly associated with ASF. These three sows were selected for sampling because they had not eaten, had elevated body temperatures, and were frequently laying down for several days.

Physiological assessments were carried out, including the measurement of body temperatures. Blood samples were also collected for comprehensive laboratory analysis. Following diagnostic tests or kits were used for on-site testing:Genome Detection: Using a portable qPCR device (Franklin^®^ Real-Time PCR Thermocycler from Biomeme, Philadelphia, PA, USA, following King et al., 2003 [23]).Antigen Detection: Using antigen lateral flow tests (INgezim ASFV CROM Ag lateral flow assay (Eurofins Technologies Ingenasa, Madrid, Spain)).Antibody Detection: Using an ELISA test (INgezim PPA COMPAC- Blocking ELISA, Eurofins Technologies Ingenasa, Madrid, Spain, for IgG and IgM detection).

All used tests were certified, commercially available products performed according to the manufacturer’s instructions.

## 3. Results and Discussion

The three animals, tested on-site, were qPCR-positive, negative in the rapid antigen test, and positive in the antibody test (Table 1). The presence of antibodies indicates that the sows had been infected with ASF for more than two weeks, which explains the negative antigen test results and comparatively high ct-values (Table 1). It should be noted that the negative antigen result may also be due to low test sensitivity [24,25]. These findings suggest that the sows had been harboring the virus for at least two weeks while remaining within the stable environment. During this period, there was a potential for the animals to shed the virus, posing a risk of transmission to other animals. Therefore, it is critical to implement strategies that significantly shorten the duration for which infected animals can remain undetected and capable of spreading the virus within a herd. Partial culling, if not implemented correctly, can carry great risks. The presence of undetected infected animals within the farm can lead to prolonged viral shedding and increased opportunities for transmission. Without rigorous testing and immediate removal of suspected cases, the virus can persist and spread, undermining the effectiveness of disease control efforts.

Under optimal conditions for disease control, one would proceed as follows: complete culling of all animals within an ASF-affected epidemiological unit and potentially across the entire farm. Detailed epidemiological investigations would then be conducted to ascertain the time and route of ASF introduction and to identify contact holdings where the virus could have spread. These investigations would be supported by a comprehensive diagnostic testing program, ensuring accurate data on the virus’s spread and persistence within the farm and broader affected region. However, the implementation of such an optimal control strategy is seldom feasible in field conditions, primarily due to constraints in financial and human resources. Additionally, the obscure historical data of a disease outbreak often leads to reliance on assumptions rather than facts, complicating the identification of index cases [26]. In the present example, this information is notably absent. Determinations regarding the temporal onset of signs, mortality dates, and the extent of diagnostic testing remain elusive. This scenario typifies the informational deficiencies encountered in field settings attributable to a variety of undisclosed factors. Under these challenging conditions, this example serves as a framework for exploring how disease management can be enhanced in environments constrained by limited diagnostic resources and changing clinical courses. It is important to emphasize that partial or selective culling is not a scientifically proven or internationally accepted containment strategy for ASF. Instead, this manuscript aims to evaluate the consequences of such practices and highlight their potential risks.

In response to an outbreak of ASF, culling strategies can be categorized into three primary methods: (i) total culling, also known as “stamping out”, where all pigs within an infected holding are culled; (ii) partial culling, targeting only pigs within a specifically infected epidemiological unit; and (iii) selective culling, where only those pigs exhibiting signs or suspected of infection are culled (Figure 5) [9,22]. The stamping-out approach, combined with standstill measures, represents the conventional control measure adopted by many countries. However, the emergence of ASF in previously unaffected countries often places an overwhelming burden on veterinary authorities due to the intense resources required for implementing total culling strategies. Challenges in executing such strategies include significant organizational demands, extensive use of human and financial resources, and profound impacts on the livelihoods of farmers with a lack of compensation. These impacts are compounded by ethical concerns regarding the culling of healthy animals [4,27,28]. In Vietnam, following the widespread transmission of ASF, regulations have been adapted to permit partial and selective culling strategies. This adaptation allows for the retention of pigs that test negative for ASF, helping to conserve resources, mitigate the environmental consequences of mass carcass disposal, and preserve the economic stability of farmers’ operations. Such measures reflect a shift towards more sustainable and ethically considerate management practices in the face of continuing ASF challenges. 

Partial culling emerges as a viable strategy when ASF infection is detected early and confined to a few epidemiological units, such as individual stables on a farm. However, this strategy becomes impractical when the infection permeates most epidemiological units across the farm, necessitating a shift to selective culling, often described as the “tooth extraction” strategy. This approach is particularly advantageous in sow farms where animals are housed individually, enhancing its effectiveness compared to fattening farms where animals are grouped together. The rationale behind selective culling is to minimize the number of animals removed, aiming to preserve valuable breeding sows, which represent significant investments for farmers. Despite its intuitive appeal, selective culling carries the risk of allowing the virus to persist within the farm if not meticulously managed. 

The persistence of infected animals, even for short periods, can lead to extensive viral shedding and environmental contamination as virus excretion intensifies during the clinical phase of the disease. Research indicates that viral loads peak in bodily fluids and excretions in the days leading up to an animal’s demise [29]. Consequently, it is crucial to identify and remove infected animals at the onset of signs, when viral shedding is relatively low, to mitigate the spread of the virus. To ensure effective disease management, early diagnostic testing of suspected cases is essential. This enables the confirmation of ASF infection at an early stage, allowing for timely intervention that curtails further viral transmission. Implementing rigorous early testing protocols could substantially decrease environmental contamination and prevent the infection of animals with direct or secondary contact, thereby enhancing the overall effectiveness of disease control measures on the farm. 

Approximately 50% of the F1 pigs became infected and either succumbed to the disease or had to be culled. This underscores a primary objective of the selective culling strategy: to reduce the incidence of infections. A practical and effective approach to achieving this goal involves the early removal of sick, infected animals from the herd. In our study, the three sick F1 animals tested positive for ASF had been ill for a period ranging from 4 to 14 days, likely shedding the virus throughout this time. Such prolonged viral shedding creates numerous opportunities for virus transmission within the herd. Reducing this exposure time is crucial, as it would lower the infection rate among animals and ultimately reduce overall mortality, aligning with the farmer’s interests. Although the selective culling approach implemented on the farm did not completely eradicate ASF, the farmer believed it contributed to preventing a significant spike in mortality. The mortality rate remained stable, with up to five sows lost per day during the second wave of infection. However, despite these efforts, the outcomes were still not satisfactory, indicating the need for further refinement and optimization of selective culling protocols to achieve better control of ASF.

The most reliable way of detecting an infected animal at an early stage, for example, at the first sign of disease, is to test the animal for the ASF virus genome (qPCR). Such examinations were unfortunately not carried out regularly; decisions to cull were based mainly on the clinical picture, namely on the onset of severe signs of the disease. There are several reasons why ASF testing was not done. On the one hand, qPCR tests and antigen tests are relatively expensive, apart from their availability, and on the other hand, in the case of qPCR diagnostics, the corresponding PCR machine is needed and has to be operated by a trained technician. These are surmountable obstacles in practice. For selective culling to be effective, qPCR or antigen tests must be carried out directly on the farm so that a decision can be made on the same day whether or not to remove an animal. If a farmer is not able to carry out regular pen-side tests to detect an infected animal at an early stage, he can still assume that sick animals, e.g., those that have not eaten for a few days, are infected, and he could remove them from the herd. However, there is a risk that animals that are not infected with the ASF virus and have fallen ill for completely different reasons will also be removed. Nevertheless, this risk should be negligible on a farm that is endemically infected with ASF, and selective culling is implemented. However, even on farms where sick animals cannot be tested continuously, qPCR tests should be carried out at monthly intervals by an accredited laboratory to monitor the ASF incidence.

Experimental evidence and recent field studies have shown that ASF is a less contagious disease and that virus transmission between animals and farms is more of a delayed process [30,31,32,33]. The relatively slow spread of the virus was also observed on the farm studied. None of the F2 animals on the farm became infected. As a result, control measures such as selective culling can be effective if applied appropriately. The Swine Health Information Center (SHIC) conducted in 2021 a study, suggesting that “tooth extraction” was not sufficient to eliminate ASF from a sow farm in Vietnam [34]. In fact, when the F0 sow and the two contact sows F1 on each side of her were removed, there was still a 50% probability that additional, undetected ASFV qPCR-positive sows remained among the F2 animals. An important difference is that sows were housed continuously next to each other, whereas in this case, one cage was left free after every three animals, possibly explaining the increased number of ASFV-positive F2 animals in the experiments conducted by SHIC. It has been shown that direct contact between sick animals is the probable transmission route [35]. Our results, along with those from SHIC, indicate that spacing animals one cage apart effectively cut transmission to F2. The primary transmission routes—direct contact and oral uptake—make the disease less contagious than other animal diseases, such as Foot and Mouth Disease or Classical swine fever [36]. Furthermore, spacing emerges as a cost-effective control measure, particularly feasible for low-resource regions, highlighting space as a crucial factor.

In recent years, there have been increasing reports of milder ASF strains circulating in Vietnam alongside the genotype II ASF virus, with the latter leading to severe infection with a high case fatality rate [37]. The circulation of such strains would make the control of ASF much more difficult since unnoticed spread will become more efficient. The effectiveness of selective culling would also decrease, as infected animals would be detected much later due to the rather mild and “unclear” clinical signs. A much stronger diagnostic effort would be necessary. In this case study, the focus was not on biosecurity aspects. However, there is no question that biosecurity must go hand in hand with all other control measures, including the separation of epidemiological units [33,38]. In particular, the biosecurity measures must prevent the ASF virus from being spread to other farms as alternative control strategies, e.g., selective culling, will only be widely accepted if it does not come along with increases in secondary outbreaks.

To conclude, early detection and consistent removal of infected pigs are essential when implementing selective culling. Separation of pigs to reduce contacts between animals as much as possible plays a key role as well. The best way to achieve this is to promptly test conspicuous animals and separate them. Alternatively, in the absence of testing opportunities, suspect animals should be removed from the epidemiological unit immediately. If they are not removed immediately, they should stay separated until tested by qPCR. Under no circumstances should suspect animals remain within the herd for extended periods, as delayed removal until they exhibit severe illness or death can exacerbate the spread of the disease. It is important to consider the potential drawbacks of partial culling. Without the implementation of an effective biosecurity protocol, partial culling alone is unlikely to succeed. Farms with inadequate biosecurity measures and insufficient diagnostic capabilities, such as the inability to conduct on-site tests for suspected cases, may remain endemic for extended periods and contribute to the persistence of the ASF virus. At best, ASF-related mortality on such farms remains consistently high.

## Figures and Tables

**Figure 1 pathogens-13-00567-f001:**
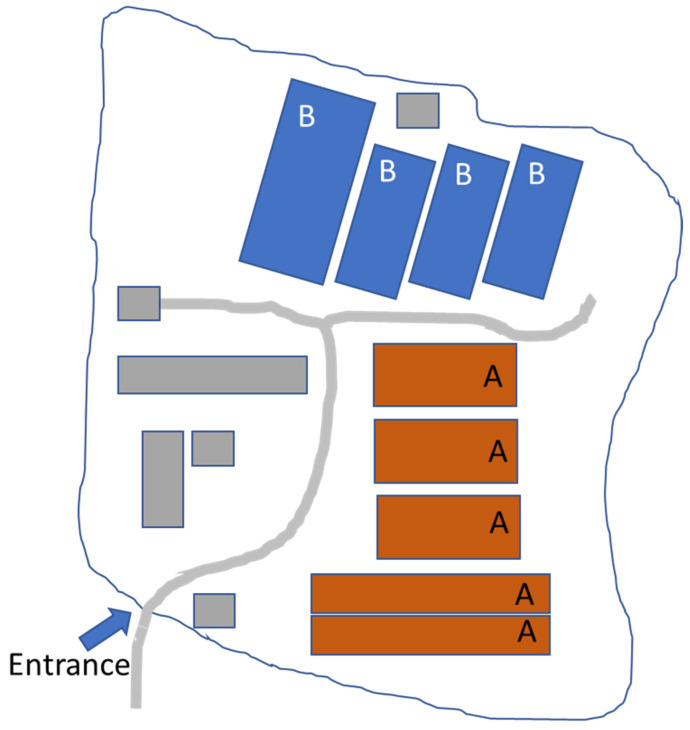
Schematic View of the Farm Layout and Structure.

**Figure 2 pathogens-13-00567-f002:**
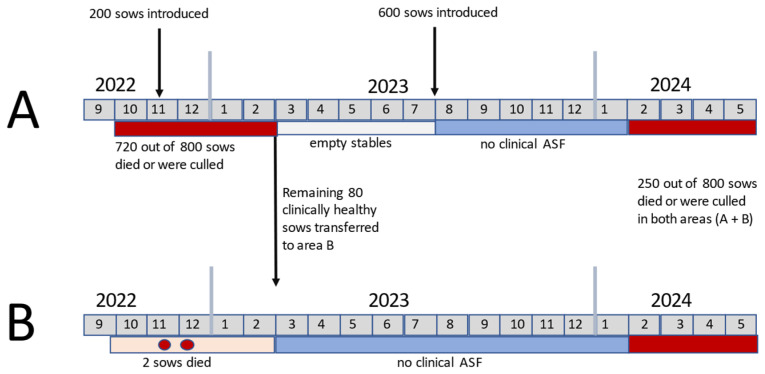
Timeline of ASF infection in both sections (**A**,**B**) of the sow farm. Red bars indicate clinical infections while blue bars indicate no clinic in pigs.

**Figure 3 pathogens-13-00567-f003:**
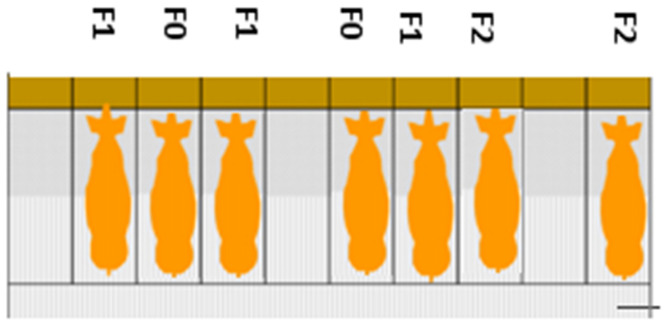
Housing of sows on an ASF-infected farm: F0—sows with signs of infection; F1—sows in direct contact with F0; F2—sows in direct contact with F1 or in the same row as F1.

**Figure 4 pathogens-13-00567-f004:**
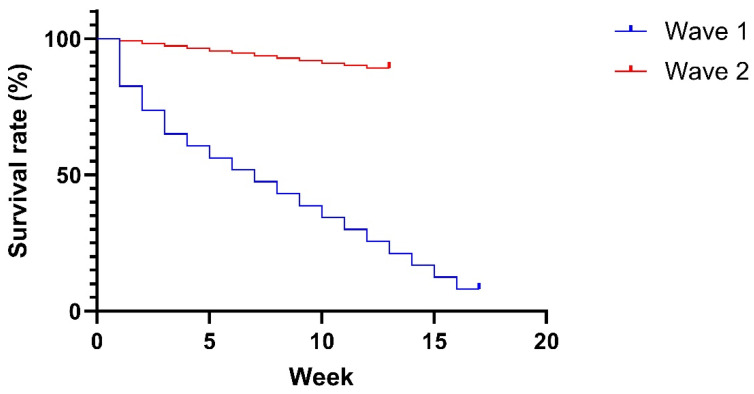
Kaplan–Meier survival curve comparing the survival rates of the two African swine fever waves on a sow farm in Vietnam.

**Figure 5 pathogens-13-00567-f005:**
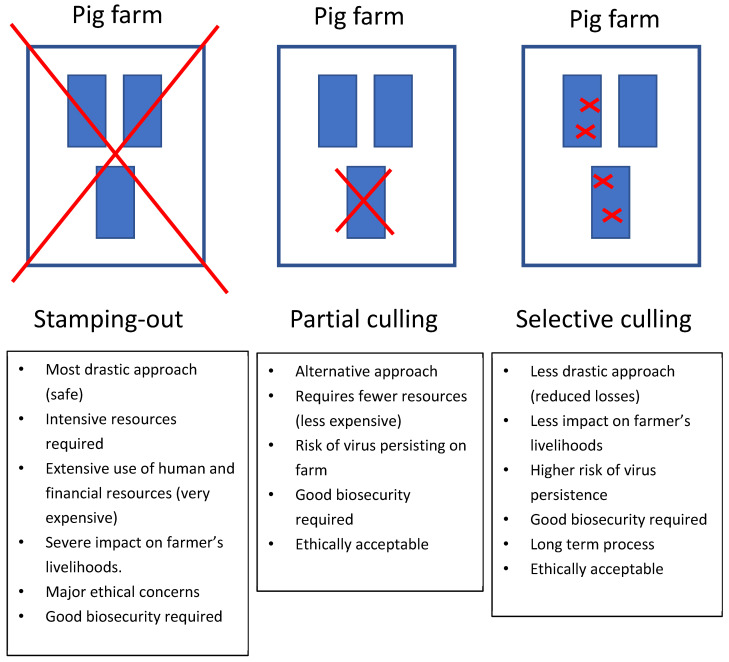
Pros and cons of stamping-out, partial culling and selective culling.

**Table 1 pathogens-13-00567-t001:** Clinical and Laboratory Results of Three Sows Tested for ASF.

Sow	Days Since Stopped Eating	Body Temperature °C	qPCR (Ct-Value)	AG-LFD	Ab-Elisa
1 (7879)	4	38.03	pos (28)	neg	pos
2 (7973)	14	38.5	pos (24)	neg	pos
3 (8008)	13	38.0	pos (30)	neg	pos

## Data Availability

Data is available upon request.

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
