# Peer review of "Evaluation of Selective Culling as a Containment Strategy for African Swine Fever at a Vietnamese Sow Farm"

_pathogens, 2024, doi:10.3390/pathogens13070567_

Round 1
Reviewer 1 Report
Comments and Suggestions for Authors
Introduction
Line 30: “Suidae” need to be in italic
Line 34-35: please provide here reference
Line 41: 4-10 pigs – are they small scale farms or backyards? Please clarify
Line 53: this official letter 5169/BNN-TY should be also in the list of references
Material and method
Line 75: it is not clear how farm with a capacity of 1200 sows can be housed in two areas? Is it divided according to production phases? And part A and B are only 50 meters apart? So they are like one farm with different stables or not?
This part where describing the farm as an epidemiological unit is really important. It is suggested to create a schematic view of the farm or exact plan of the farm, if available, to make it clear to the reader how the objects are placed. The given description is confusing: how farm can be located in two areas which are only 50 meters apart? What kind of areas are there? Is it fenced area? Is it multi-site production system or farrow-to-finish? It is necessary to correct this part in the manuscript.
Line 110: direct neighbours= according to the fig.2 they are in direct contact. So, word “neighbours” should be avoided. The animals are placed in direct contact, which is very important for ASF spreading and transmission.
Line 125: please correct “areas”
Line 128-129: don’t you think that fewer also belong to this list of clinical symptoms?
Results and discussion
Line 197-200: please provide the reference at the end of the sentence
Line 206: what is the epidemiological unit according to this paper/according to the law in this country? This could be interesting to add in discussion
Figure 5: in all these good biosecurity is required (please add to “stamping out”)
Line 232: who are the neighbouring animals? A neighbour is someone who is living next to your house. But are these animals in the same stable? In that case they are not neighbouring. If they are in the same stable, next to each other they are in direct contact.
References
The reviewer cannot find the references number 23-26 in the final list. They are in the part of results and discussion but not in the list of references.
Author Response
Dear Reviewer,
Thank you for your detailed feedback and suggestions. Here are our responses to your comments:
Comments 1: Line 30: “Suidae” needs to be in italics.
Response 1: This is changed accordingly.
Comments 2: Line 34-35: Please provide a reference here.
Response 2: Thank you for pointing this out. We have added the necessary references as indicated in the revised manuscript.
Comments 3: Line 41: 4-10 pigs – are they small scale farms or backyards? Please clarify.
Response 3: Thank you for pointing this out. We have clarified that the farms are backyard farms.
Comments 4: Line 53: This official letter 5169/BNN-TY should also be in the list of references.
Response 4: The reference was added: Ministry of Agriculture and Rural Development. (2019, July 22). Công văn 5169/BNN-TY 2019 hướng dẫn bổ sung biện pháp phòng chống bệnh Dịch tả lợn Châu Phi. Thư Viện Pháp Luáºt.
Comments 5: Line 75: It is not clear how a farm with a capacity of 1200 sows can be housed in two areas. Is it divided according to production phases? And part A and B are only 50 meters apart? So they are like one farm with different stables or not? This part where describing the farm as an epidemiological unit is really important. It is suggested to create a schematic view of the farm or exact plan of the farm, if available, to make it clear to the reader how the objects are placed. The given description is confusing: how can the farm be located in two areas which are only 50 meters apart? What kind of areas are there? Is it a fenced area? Is it a multi-site production system or farrow-to-finish? It is necessary to correct this part in the manuscript.
Response 5: A map was included in the manuscript and necessary information added (Materials and Methods Section-Farm description and Figure 1).
Comments 6: Line 110: direct neighbours= according to Fig.2 they are in direct contact. So, the word “neighbours” should be avoided. The animals are placed in direct contact, which is very important for ASF spreading and transmission.
Response 6: This was changed accordingly.
Comments 7: Line 125: please correct “areas”.
Response 7: Areas is changed to sections.
Comments 8: Line 128-129: Don’t you think that fever also belongs to this list of clinical symptoms?
Response 8: Fever was added into the list.
Comments 9: Line 197-200: Please provide the reference at the end of the sentence.
Response 9: References were added.
Comments 10: Line 206: What is the epidemiological unit according to this paper/according to the law in this country? This could be interesting to add in the discussion.
Response 10: An epidemiological unit, as described in the manuscript, refers to a defined group of animals within a farm that are managed together and share the same risk of exposure to a disease. Each section of the farm (A and B) operates as an independent epidemiological unit. Animals in different units do not mix, except during the initial quarantine period, to minimize the risk of disease spread. This structure allows for effective disease control measures, such as selective culling, to be implemented within each unit based on the health status and risk factors specific to that group. There is no definition by Vietnam. A paragraph was added into Materials and Methods-Farm description.
Comments 11: Figure 5: In all these, good biosecurity is required (please add to “stamping out”).
Response 11: This was added accordingly.
Comments 12: Line 232: Who are the neighboring animals? A neighbor is someone who is living next to your house. But are these animals in the same stable? In that case, they are not neighboring. If they are in the same stable, next to each other, they are in direct contact.
Response 12: This was changed accordingly.
Comments 13: References: The reviewer cannot find the references number 23-26 in the final list. They are in the part of results and discussion but not in the list of references.
Response 13: The missing refs are included.
Thank you again for your thorough review and helpful suggestions.
Best regards,
Dr. Auer
Reviewer 2 Report
Comments and Suggestions for Authors
Dear Authors,
It is great to see this study to publication in the journal. The content of the paper is straightforward to study. Table #1 needs to change the correct format to publication, such as 38,3 to 38.3; 38,5 to 38.0; 38,0 to 38.0
Overall, I agree to release the paper to the public in the journal.
Thank you,
Author Response
Dear Reviewer,
Thank you for your feedback and for bringing this to our attention. We have made the necessary changes to Table #1, ensuring the format is corrected as suggested: 38,3 to 38.3; 38,5 to 38.5; and 38,0 to 38.0.
We appreciate your thorough review and are glad to hear that you support the release of our paper in the journal.
Thank you for your assistance.
Best regards,
Reviewer 3 Report
Comments and Suggestions for Authors
The manuscript entitled "Optimizing Selective Culling as a Containment Strategy for Af-2 rican Swine Fever at a Vietnamese Sow Farm" is an short communicate, as a result from the observation of the ASFV spread in a sow breeding farm in Vietnam, with an estimated capacity of 1200 sows.
The abstract of the work promises an interesting research offer, some controversy is triggered at the very beginning of "methods" where the "approach to tooth extraction" model is later introduced (Fig. 2). This thesis seems to be confirmed by the further part of the manuscript, which states that the ASF positive animals are located at a short distance from other pigs (within one empty housing unit) with a (theoretically) healthy condition. The authors of the manuscript rightly point out that the determinant of many diseases (including ASF) is the distance that determines the possibility of their transmission.
The issue exists beyond doubt that observations made in Vietnam, there are "approaches to tooth extraction" when it exists, which in the event of the patient's comment raises fundamental concerns. It is common to say that ASF affects not only wild boars, but primarily large numbers of domestic pigs, sometimes grouped in huge holdings. An essential element of the fight against ASF is the elimination of the entire population that could potentially have been in contact with the ASF virus and this issue is not subject to general discussion. There is too great a risk that a potentially healthy pig, staying, as the authors of the paper suggest, two cages away from a sick pig, could be infected with the ASF virus, but due to too early symptoms, this disease would not be properly detected, and the tissues of this animal could be lead to further transmission of ASF disease. ASF in the natural environment, and even more so in pig breeding farms, can be transmitted very quickly, as evidenced by numerous publications, but the analysis and verification of the animals' health status often requires time needed to induce ASF symptoms and an immune response. I have the impression that these factors should be re-analyzed by the authors of the work, with particular emphasis on numerous literature data on the transmission of ASFV and the possibility of its transmission, for example through air circulation in a livestock building. Without the doubts, the observations made in Vietnam, related to "tooth extraction approach" are interesting, but raises fundamental concerns when compared with many data taken from scientific articles. It is common to say that ASF affects not only wild boars, but primarily large numbers of domestic pigs, sometimes grouped in huge farm holdings. An essential element of the fight against ASF is the elimination of the entire population that could potentially have been in contact with the ASF virus and this issue is not the subject to general discussion. There is too high risk that a potentially healthy pig, staying, as the authors of this paper suggest, two cages away from a sick pig, could be infected with the ASF virus. What is important, early symptoms, may not be detected properly what may be the result of next ASF spread of this disease. In my opinion the authors should consider their opinion as well as precisely (but in comparison with laboratory data) determine the premises for possible further conclusions from mentioned observations.
I strongly recommend these positions:
1) https://www.woah.org/app/uploads/2022/07/asf-in-wild-boar-ecology-and-biosecurity-2nd-ed.pdf
2) https://www.mdpi.com/1999-4915/11/4/310
3) https://www.mdpi.com/2076-0817/10/2/177
4) https://www.mdpi.com/2076-0817/9/12/1077
5) https://www.mdpi.com/1999-4915/15/11/2275
Additional confusing data were pointed here:
line: 128-132 "Sick sows were defined as having ASF if they had not eaten for several days and were lying down, unable to get up. In cases of milder signs like loss of appetite, animals were first treated with antibiotics to prevent secondary infections, although it is known that this does not combat ASF itself. Culling was based solely on clinical signs since ASF-specific laboratory tests were not conducted"
- why the antibiotics were used here? there was any specific reason of it?
- if yes, this animals should be removed from the experiment!
- if no - any justification of its use should be given here!
line 187-191 "In response to an outbreak of ASF, culling strategies can be categorized into three primary methods: (i) total culling, also known as "stamping out," where all pigs within an infected holding are culled; (ii) partial culling, targeting only pigs within a specifically infected epidemiological unit; and (iii) selective culling, where only those pigs exhibiting signs or suspected of infection are culled (Fig. 4)".
- do you have any scientific conformation of these sentences?
- when and where the "partial culling" and "selective culling" were effective to stop the spread of the ASFV?
This manuscript is interesting, it is even possible that these propositions could be used in the future in the fight against ASF. At the moment, in the light of the knowledge of ASF, both the methods and the conclusions cannot be considered as appropriate and correct. In the opinion of the reviewer, the authors should reconsider the research concept and verify the content of this manuscript before sending this text to publisher.
Author Response
Dear Reviewer,
Thank you for the feedback on our manuscript. It has certainly helped us uncover some misunderstandings and ambiguities, and we have addressed them accordingly.
An important and valid argument of the reviewer is: “An essential element of the fight against ASF is the elimination of the entire population that could potentially have been in contact with the ASF virus...” We completely agree with the reviewer. By culling all susceptible animals, the disease can be successfully eliminated. This approach is legally required in the European Union and is also the method of choice in wealthy countries. However, this drastic approach presents a significant dilemma for less wealthy and developed countries. These countries lack the financial and human resources to cull an entire population that could be infected (leaving aside ethical arguments). This is only feasible in countries where the state provides 100% compensation to the farmer. In places where the state cannot afford this, total stamping out (at the farmer's expense) is simply unrealistic. No ASF-affected country in Asia, including Vietnam, can consistently implement total culling, even though it is scientifically the preferred method. Our paper aims to show how alternative procedures to stamping out (though scientifically the best method) could be optimized and improved in a country like Vietnam, where total stamping out has proven impossible. The reasons why total stamping out, which is favored in the EU, cannot be implemented in a country like Vietnam have already been discussed in a scientific paper (Nga, et al 2022. https://doi.org/10.3389/fvets.2022.957918) .
In addition, we have included a final paragraph to the conclusion: 'It is important to consider the potential drawbacks of partial culling. Without the implementation of an effective biosecurity protocol, partial culling alone is unlikely to succeed. Farms with inadequate biosecurity measures and insufficient diagnostic capabilities, such as the inability to conduct on-site tests for suspected cases, may remain endemic for extended periods and contribute to the persistence of the ASF virus. At best, ASF-related mortality on such farms remains consistently high.'
The reviewer has asked three questions about the text in lines 128 - 132.
line: 128-132 "Sick sows were defined as having ASF if they had not eaten for several days and were lying down, unable to get up. In cases of milder signs like loss of appetite, animals were first treated with antibiotics to prevent secondary infections, although it is known that this does not combat ASF itself. Culling was based solely on clinical signs since ASF-specific laboratory tests were not conducted"
- why the antibiotics were used here? there was any specific reason of it?
- if yes, this animals should be removed from the experiment!
- if no - any justification of its use should be given here!
There seems to be a misunderstanding here. In our paper, we do not describe an experiment or a field study but report on a case as we found it in the field. Regarding the use of antibiotics, it is unfortunately common practice, not only in Asia and Vietnam but also in Europe (EU), to treat animals with antibiotics at the first sign of illness in the hope that the treatment will be effective and that the animals are not infected with a viral disease such as ASF. The answer to why antibiotics were used is that it is generally accepted practice, even though it is scientifically incorrect. Unfortunately, this is the reality that must be recognized and taken into account. We do not justify incorrect human behavior here; we only describe what actually took place.
The reviewer has asked two questions about the text in lines 187 - 191.
line 187-191 "In response to an outbreak of ASF, culling strategies can be categorized into three primary methods: (i) total culling, also known as "stamping out," where all pigs within an infected holding are culled; (ii) partial culling, targeting only pigs within a specifically infected epidemiological unit; and (iii) selective culling, where only those pigs exhibiting signs or suspected of infection are culled (Fig. 4)".
- do you have any scientific conformation of these sentences?
- when and where the "partial culling" and "selective culling" were effective to stop the spread of the ASFV?
Within the EU, only total stamping out is implemented. However, outside the EU, different culling methods are a reality (see above), and clear definitions are needed to better understand what is happening and where strategies are failing. As ASF has a major impact in Asia, where stamping out is not an option, it will be helpful to gain and share more experience with partial or selective culling strategies. Culling animals in response to an outbreak is a legally prescribed animal disease control measure. Therefore, the competent authority, which must carry out the control measures, primarily needs a legal basis to do so, not just scientific justification. However, scientific justification underpins the legislator's decision to draft a corresponding legal act. In this case, we have listed the culling methods used worldwide, in line with the legal requirements where they are employed. For example, total stamping out and partial culling are clearly regulated in the EU Animal Health Law, and the Vietnamese Ministry of Agriculture has issued corresponding regulations for selective culling. We are not aware of any other culling applications beyond these three, nor any legal or scientific considerations. To justify this, we have referred to the legal requirements in line 211.
We hope that we have been able to answer the reviewer's questions with these explanations.
The reviewer is strongly recommended to include the following references:
1) https://www.woah.org/app/uploads/2022/07/asf-in-wild-boar-ecology-and-biosecurity-2nd-ed.pdf
2) https://www.mdpi.com/1999-4915/11/4/310
3) https://www.mdpi.com/2076-0817/10/2/177
4) https://www.mdpi.com/2076-0817/9/12/1077
5) https://www.mdpi.com/1999-4915/15/11/2275
We have included the references to our manuscript.
We appreciate your thorough review and constructive comments, which have helped improve the clarity and quality of our manuscript. Thank you for your valuable input.
Kind regards,
Dr. Auer
Round 2
Reviewer 3 Report
Comments and Suggestions for Authors
The second version of the manuscript "Optimizing Selective Culling as a Containment Strategy for African Swine Fever at a Vietnamese Sow Farm" was updated. Maintaining the reviewer's theses contained in the first review, I would like to indicate my attitude to the Authors' answers:
1) it was said "This is only feasible in countries where the state provides 100% compensation to the farmer. In places where the state cannot afford this, total stamping out (at the farmer's expense) is simply unrealistic."
- reviewer's opinion: The authors can collect data on animal production failures in Vietnam, especially regarding the prevention and control of ASFV. However, this does not change the fact that in scientific studies, this type of breeding practices should be subject to deep criticism and strong denial. Meanwhile, the authors, through this type of publications, may cause "relaxed" treatment of the ASF disease, which will absolutely not help breeders in Vietnam - on the contrary - it may lead to a greater spread of ASF in this country. I find these explanations as unsatisfactory.
2) it was said "The answer to why antibiotics were used is that it is generally accepted practice, even though it is scientifically incorrect. Unfortunately, this is the reality that must be recognized and taken into account."
- reviewer's opinion: This publication would appear in a respected, peer-reviewed journal with a wide range of impact. Therefore, since it is common knowledge that antibiotics are not intended to prevent the spread of viral diseases, it should be expected that the content of scientific articles in international journals will strongly condemn this type of practice. This element is definitely missing here and even if it is a "generally accepted practice" in Vietnam, it is necessary to point out that "even though it is scientifically incorrect".
3) it was said: "Therefore, the competent authority, which must carry out the control measures, primarily needs a legal basis to do so, not just scientific justification. However, scientific justification underpins the legislator's decision to draft a corresponding legal act."
- reviewer's opinion: The reviewer again believes that promoting scientifically unjustified practices can only strengthen the position of ASF with further spread of the disease and will not contribute to its elimination. Comparing the legislative rules in Vietnam and EU countries could be right, but in European countries, legal acts regarding ASF were firstly created based on objective scientific opinions (inclouding EFSA). In that case, the reviewer has the impression that this publication may be important to give a scientific basis for the political decisions in Vietnam, which simply (if the option of "tooth extraction approach" is going to be used) may pose a risk of further spread of ASFV, which cannot be allowed. Moreover, there is still lack of the answers made lastly to the line 187-191: do you have any scientific conformation of these sentences? when and where the "partial culling" and "selective culling" were effective to stop the spread of the ASFV?
To sum up, I appreciate the authors' contribution in the manuscript preparation. Nevertheless, I my opinion is that its content is not adequate to the current state of knowledge in the field of ASF elimination, and this publication may lead to the spread of this dangerous disease. Considering the above, I do not recommend this text for publication in Pathogens.
Author Response
- Comment: The second version of the manuscript "Optimizing Selective Culling as a Containment Strategy for African Swine Fever at a Vietnamese Sow Farm" was updated. Maintaining the reviewer's theses contained in the first review, I would like to indicate my attitude to the Authors' answers: 1) it was said "This is only feasible in countries where the state provides 100% compensation to the farmer. In places where the state cannot afford this, total stamping out (at the farmer's expense) is simply unrealistic." - reviewer's opinion: The authors can collect data on animal production failures in Vietnam, especially regarding the prevention and control of ASFV. However, this does not change the fact that in scientific studies, this type of breeding practices should be subject to deep criticism and strong denial. Meanwhile, the authors, through this type of publications, may cause "relaxed" treatment of the ASF disease, which will absolutely not help breeders in Vietnam - on the contrary - it may lead to a greater spread of ASF in this country. I find these explanations as unsatisfactory.
1st answer:
Dear Reviewer,
We appreciate your thorough review and the opportunity to address your concerns. While we understand and respect your perspective, we would like to offer some additional context and evidence to support our approach.
The assertion that alternative methods (selective and partial culling) will lead to a greater spread of the disease is neither proven nor comprehensible. In fact, the number of pigs has been increasing again since the Ministry of Agriculture and Rural Development amended the ASF control policy to include the possibility of partial and selective culling on infected farms (General Statistics Office of Vietnam, n.d.). In contrast, total stamping-out has proven to be impractical and unfeasible, aside from the social and ethical concerns it raises. Unfortunately, Vietnam is currently almost entirely affected by ASF, so the argument for a larger spread is not plausible. A significant amount of data on the ASF situation and production conditions in Vietnam has already been collected and published (e.g., Nga et al. 2022). In addition, Vietnam is the first and, so far, only country where an ASF vaccine has been developed and authorized, with numerous publications resulting from this work. Mentioning the development and use of vaccines is crucial because it highlights an essential component of the multi-faceted approach needed to control ASF in regions where conventional methods, such as total stamping-out, are not feasible. At the beginning of the epidemic in Asia (2018-2019), the total stamping-out method, which you advocate, was used extensively. However, this approach ultimately led to a massive spread of the disease in Asia rather than containing it. In fact, as result of the implemented control strategies, ASF has to be considered largely endemic now.
We respectfully disagree with the characterization of selective or partial culling as a "relaxed" strategy. Rather, it is an alternative born out of necessity. In countries like Vietnam, total culling is often unfeasible due to financial, logistical, and ethical constraints. Consequently, alternatives to total stamping-out, including selective and partial culling, are being tested and implemented. These alternatives aim to control the disease while considering the practical limitations faced by farmers and authorities. Therefore, we have made major changes to the text, emphasizing that selective culling harbours great risks if not implemented correctly and that, so far, there is no scientifically approved protocol at hand. As such, this manuscript evaluates the consequences of these practices as currently applied in Vietnam and points out the potential risks. The issue with alternative methods (partial and selective culling) lies in their implementation without a scientific basis. This is the point we address in our work. Our goal is to ensure that when partial or selective culling is used, it is based on scientific evidence.
- Comment: it was said "The answer to why antibiotics were used is that it is generally accepted practice, even though it is scientifically incorrect. Unfortunately, this is the reality that must be recognized and taken into account."
- reviewer's opinion: This publication would appear in a respected, peer-reviewed journal with a wide range of impact. Therefore, since it is common knowledge that antibiotics are not intended to prevent the spread of viral diseases, it should be expected that the content of scientific articles in international journals will strongly condemn this type of practice. This element is definitely missing here and even if it is a "generally accepted practice" in Vietnam, it is necessary to point out that "even though it is scientifically incorrect".
2nd Answer:
Here we agree with the reviewer that the uncontrolled use of antibiotics must be criticised and condemned even more strongly. We included this point in the manuscript accordingly: ’The authors would like to emphasize, that antibiotics are not intended to prevent the spread of viral diseases and such practices on an ASF infected farm should not be followed. However, this was the practice on this farm visited.’ Although it was already stated in lines 157 that antibiotics do not treat viral diseases: ‘In cases of milder signs like loss of appetite, animals were first treated with antibiotics to prevent secondary infections, although it is known that this does not combat ASF itself.’
- Comment: it was said: "Therefore, the competent authority, which must carry out the control measures, primarily needs a legal basis to do so, not just scientific justification. However, scientific justification underpins the legislator's decision to draft a corresponding legal act."
- reviewer's opinion: The reviewer again believes that promoting scientifically unjustified practices can only strengthen the position of ASF with further spread of the disease and will not contribute to its elimination. Comparing the legislative rules in Vietnam and EU countries could be right, but in European countries, legal acts regarding ASF were firstly created based on objective scientific opinions (including EFSA). In that case, the reviewer has the impression that this publication may be important to give a scientific basis for the political decisions in Vietnam, which simply (if the option of "tooth extraction approach" is going to be used) may pose a risk of further spread of ASFV, which cannot be allowed. Moreover, there is still lack of the answers made lastly to the line 187-191: do you have any scientific conformation of these sentences? when and where the "partial culling" and "selective culling" were effective to stop the spread of the ASFV?
3rd Answer:
We would like to address your concerns, particularly regarding ASF European legislation and its scientific basis. Firstly, we believe there might be a misunderstanding regarding the principles of ASF control measures. The recitals of the repealed Council Directive 2002/60/EC or the AHL clearly indicate that the principles of ASF control measures in ASF legal acts are derived from previous CSF legislation. Consequently, ASF control largely follows the measures used to control CSF, a less contagious disease controlled similarly to a highly contagious one (for details see Lamberga et al. 2024 and the recitals of the repealed Council Directive 2002/60/EC). It is also crucial to recognize that all EU strategies and policies are evolving documents. For example, while HPAI vaccination was previously banned in the EU, this policy has changed due to the recognition of "endemicity."
Regarding the objection that there are no scientific arguments in favor of selective or partial culling, we refer to the publication by Nga et al.2022, which scientifically proves that alternative culling methods have had a favorable effect on limiting disease spread and incidence compared to total stamping out. Similarly, in South Africa, a partial culling strategy successfully eradicated ASF in a specific area (van Rensburg et al., 2020).
In an ASF endemic context, which currently prevails in Vietnam and many other Asian and African countries, discussing eradication is unrealistic. According to basic disease control principles, in an endemic scenario, the incidence must first be greatly reduced before aiming for complete eradication. Total stamping out might be the method of choice in the final stage, but we are far from that point, potentially years or even decades away, not just in Asia but globally. This, unfortunately, is the current reality.
Considering your feedback, we have revised the title, abstract, and text to clearly convey the aim of this manuscript. The purpose of this manuscript is to evaluate the consequences of selective culling as it is currently implemented in Vietnam. It is important to emphasize that partial or selective culling is neither a scientifically proven nor an internationally accepted containment strategy for ASF. Instead, this manuscript aims to assess the consequences of these practices and highlight their potential risks (lines 74, 197,215, 339).
Thank you for considering our responses and for your ongoing efforts to improve our manuscript.
Dr. Auer
Reference:
General Statistics Office of Vietnam. (n.d.). Agriculture, Forestry and Fishing. Retrieved June 27, 2024, from PX Web – General Statistics Office of Vietnam (gso.gov.vn).
Nga, BTT, Padungtod, P, Depner, K, Chuong, VD, Duy, DT, Anh, ND, et al. Implications of partial culling on African swine fever control effectiveness in Vietnam. Front Vet Sci. (2022) 9:957918. doi: 10.3389/fvets.2022.957918.
Janse van Rensburg, L.; Van Heerden, J.; Penrith, M. L.; Heath, L. E.; Rametse, T.; Etter, E. M. C. Investigation of African Swine Fever Outbreaks in Pigs Outside the Controlled Areas of South Africa, 2012-2017. J. S. Afr. Vet. Assoc. 2020, 91 (0), e1-e9. DOI: 10.4102/jsava.v91i0.1997.
